# The Role of Maresins in Inflammatory Pain: Function of Macrophages in Wound Regeneration

**DOI:** 10.3390/ijms20235849

**Published:** 2019-11-21

**Authors:** Sung-Min Hwang, Gehoon Chung, Yong Ho Kim, Chul-Kyu Park

**Affiliations:** 1Gachon Pain Center and Department of Physiology, College of Medicine, Gachon University, Incheon 21999, Korea; unclehwang76@gmail.com; 2Department of Oral Physiology and Program in Neurobiology, School of Dentistry, Seoul National University, Seoul 08826, Korea; gehoon@snu.ac.kr; 3Dental Research Institute, Seoul National University, Seoul 03080, Korea

**Keywords:** inflammation, macrophage, specialized pro-resolving mediators, maresin, pain

## Abstract

Although acute inflammatory responses are host-protective and generally self-limited, unresolved and delayed resolution of acute inflammation can lead to further tissue damage and chronic inflammation. The mechanism of pain induction under inflammatory conditions has been studied extensively; however, the mechanism of pain resolution is not fully understood. The resolution of inflammation is a biosynthetically active process, involving specialized pro-resolving mediators (SPMs). In particular, maresins (MaRs) are synthesized from docosahexaenoic acid (DHA) by macrophages and have anti-inflammatory and pro-resolving capacities as well as tissue regenerating and pain-relieving properties. A new class of macrophage-derived molecules—MaR conjugates in tissue regeneration (MCTRs)—has been reported to regulate phagocytosis and the repair and regeneration of damaged tissue. Macrophages not only participate in the biosynthesis of SPMs, but also play an important role in phagocytosis. They exhibit different phenotypes categorized as proinflammatory M1-like phenotypes and anti-inflammatory M2 phenotypes that mediate both harmful and protective functions, respectively. However, the signaling mechanisms underlying macrophage functions and phenotypic changes have not yet been fully established. Recent studies report that MaRs help resolve inflammatory pain by enhancing macrophage phagocytosis and shifting cytokine release to the anti-inflammatory M2 phenotypes. Consequently, this review elucidated the characteristics of MaRs and macrophages, focusing on the potent action of MaRs to enhance the M2 macrophage phenotype profiles that possess the ability to alleviate inflammatory pain.

## 1. Introduction

Inflammation is an immune response to harmful stimuli, including pathogens, damaged cells, toxic compounds, surgery, or irradiation [1]. Inflammation is characterized by swelling, heat, pain, redness, and loss of tissue function, which is caused by local immune, vascular, and inflammatory cell responses to infection or injury [2]. Inflammatory processes that include changes in vascular permeability, recruitment and accumulation of leukocytes, and release of inflammatory mediators, are important in the regeneration of injured tissues [3]. Therefore, inflammation is an essential defense mechanism for preserving health. A weak inflammatory response can lead to tissue destruction by harmful stimuli, while chronic unresolved inflammation may culminate in various pathological conditions, including cancer, fibrosis, and pain [4]. Wound regeneration promotes resolution of inflammation by restoring barrier function [5]. Neutrophils are the first circulating inflammatory cells to be recruited to the wound site [6]. Clinical observations demonstrating that leukocyte recruitment disorders and reduced neutrophil infiltration are associated with delayed wound healing indicate the importance of neutrophils for efficient wound repair [7]. Recent studies have shown that macrophages exhibit different functions during the immune response, with proinflammatory signaling occurring during the early stages of inflammation and, once inflammation is resolved, promotion of tissue regeneration at late stages [8,9].

Inflammatory pain indicates increased mechanical and thermal sensitivity due to inflammatory reactions [10]. These mechanisms have been extensively investigated over the past two decades [11,12]. The earliest factors causing inflammatory pain are lipid mediators (leukotrienes (LTs) and prostaglandins (PGs)) and proinflammatory cytokines (Tumor necrosis factor (TNF)-α and interleukin (IL)-1β) [13]. They sensitize nociceptors of the primary sensory neurons (peripheral sensitization) through modulation of ion channels including TRP channels [14,15,16]. However, our understanding of the resolution processes and mechanisms that causes inflammatory pain is limited. The acute inflammatory response is protective, evolved to repair damaged tissues and eliminate invading organisms [17,18]. This is ideally a self-limited inflammatory response that leads to a complete resolution of leukocyte infiltrates and removal of cellular debris, allowing for the return to normal homeostasis [18]. However, uncontrolled or unresolved acute inflammatory conditions can lead to chronic inflammation, causing greater tissue damage, tissue remodeling disorders, and poor tissue healing [19,20]. These conditions are known to induce the transition to chronic and maladaptive inflammatory pain [21,22] and may lead to vascular disease, metabolic syndromes, and neurological diseases [19].

In general, the resolution of acute inflammation is an active rather than a passive process that requires the biosynthesis of SPMs including lipoxins (LXs), resolvins (Rvs), protectins (PDs), and maresins (MaRs), derived from the omega-6 fatty acid arachidonic acid (AA) and the omega-3 polyunsaturated fatty acids docosahexaenoic acid (DHA) and eicosapentaenoic acid (EPA) [23]. SPMs turn off the inflammatory response by acting on distinct G-protein-coupled receptors expressed in immune cells that activate dual anti-inflammatory and pro-resolution activity in various animal models of inflammation [24,25,26]. In response to injury or infection, an acute inflammatory response involves early tissue edema and neutrophil infiltration, some of which transits to mature macrophages [27,28]. Macrophages are major repair mediators in peripheral nerve and spinal cord injuries [29] that exist in two polarization states [30]. These states are not fixed but instead change rapidly in response to the microenvironment [31,32,33]. M1 (classically activated) macrophages produce proinflammatory cytokines and promote nociceptor sensitization that can be converted into inflammatory pain, whereas M2 (alternatively activated) macrophages produce anti-inflammatory cytokines and promote wound healing [34]. Based on these functional roles, macrophages regulate the enhancement or alleviation of pain sensitivity under various conditions [32,35]. For example, M1 infiltration has been identified in pain-associated synovial tissue in models of muscle, joint, and paw inflammation [35,36]. In contrast, M1 deficiency has been reported to reduce increased proinflammatory cytokines and prevent local inflammatory pain in response to proinflammatory agents or chemotherapy-induced peripheral neuropathy [37]. The transition from M1 to M2 phenotypes—or the balance thereof—appears to be crucial for resolution associated with acute inflammatory response [13,38]. Spinal cord injury (SCI), a condition frequently associated with prolonged inflammatory pain, increases the abundance of M1 phenotype cells in the spinal cord [13]. Thus, the balance of M1/M2 macrophages plays an important role in the resolution of inflammation and inflammatory pain relief.

MaRs are believed to act as potent protective mediators of macrophage function [39,40] and promote the resolution of acute inflammation and tissue regeneration [23,41,42]. Recent studies report that MaRs promote inflammatory activity in macrophages; furthermore, the incubation of human macrophages with MaRs improves resolution by increasing phagocytosis and efferocytosis. These effects are likely due to various substances released by MaRs that alter macrophage function and possibly contribute to the resolution of inflammation. For example, biosynthesized MaRs downregulate proinflammatory cytokines, such as IL-1β, IL-6, and TNF-α, to induce inflammation resolution and tissue regeneration [43,44]. However, defective or delayed resolution causes chronic inflammation that can eventually lead to chronic inflammatory pain [20]. It has been reported that inflammatory resolution is reduced due to the following functional problems of the lipid mediator family and macrophages: (a) M1/M2 macrophage imbalance [45,46]; (b) reduced MaR or other SPM formation [27,42,47,48,49]; (c) impaired synthesis of DHA [50]; and (d) aging [2,51,52]. Therefore, MaRs may act directly or indirectly to reverse the inflammation relief deficiencies caused by these functional problems, thereby restoring normal inflammation relief function. Moreover, a new series of bioactive peptide-lipid conjugated mediators—MCTRs—are produced in the later stages of self-resolved infection [23,41] that regulate inflammation and resolution mechanisms as well as tissue regeneration [23,41]. Consequently, this review described the functional mechanisms of MaRs based on their potential ability to control macrophage activation and inflammatory resolution.

## 2. Tissue Inflammation and Regeneration

### 2.1. Tissue Inflammation

The inflammatory response after tissue damage is an important biological process that is essential for the survival of living organisms [1]. When tissues are damaged by infection, exposure to toxins, or mechanical damage, an inflammatory response is induced by damage-associated molecular patterns (DAMPs) and pathogen-associated molecular patterns (PRR) released by dead cells and invading organisms [53]. These molecules provoke a complex inflammatory response characterized by the recruitment, proliferation, and activation of various hematopoietic and non-hematopoietic cells, including neutrophils, macrophages, innate lymphoid cells, natural killer cells, B cells, T cells, fibroblasts, epithelial cells, endothelial cells, and stem cells, which together constitute the cellular response that orchestrates tissue repair [54,55,56].

### 2.2. Tissue Regeneration

When the wound healing reaction is well organized and controlled, the inflammatory response is quickly resolved and normal tissue structure is restored [57]. However, if the wound healing response is chronic or becomes dysregulated, it can lead to the development of pathological fibrosis or scars, which impair normal tissue function and may ultimately lead to organ failure and death [56]. Therefore, the wound-healing reaction must be strictly regulated. Biological processes involved in cutaneous wound healing include infiltration of inflammatory cells, fibroblast repopulation and new vessel formation, keratinocyte migration, and proliferation [54,58,59]. Although many cells are involved in tissue repair, macrophages exhibit significant regulatory activity at all repair and fibrosis stages and are critically involved in normal tissue homeostasis [60]. It is clear that monocytes and macrophages play more complex roles in tissue repair and in contributing to fibrosis and tissue regeneration [8]. Macrophages are an important source of chemokines, matrix metalloproteinases (MMPs), and other inflammatory mediators that induce early cellular response after injury [61]. Indeed, when macrophages are depleted soon after injury, the inflammatory response is often greatly reduced [8]. However, their removal can also reduce wound debris and cause less efficient repair and regeneration [56]. After the initial inflammatory phase subsides, the main macrophage population develops a wound healing phenotype characterized by the production of numerous growth factors, such as transforming growth factor beta 1 (TGF-β1), platelet-derived growth factor (PDGF), vascular endothelial growth factor alpha (VEGF-α), and insulin-like growth factor 1 (IGF-1) [5].

## 3. Specialized Pro-Resolving Mediators (SPMs)

### 3.1. Biosynthesis of SPMs

SPMs are bioactive autacoids enzymatically produced from the omega-6 fatty acids arachidonic acid (AA), eicosapentaenoic acid (EPA), and docosahexaenoic acid (DHA); they include lipoxins (LXs), resolvins (Rvs), protectins (PDs), and maresins (MaRs) generated via the action of lipoxygenases (LOXs) and cytochrome p450 (CYP450) and cyclooxygenase-2 (COX-2) enzymes usually found in different cell types residing in inflammatory environments [62,63]. Although many cell types can produce SPMs, immune cells, such as neutrophils, monocytes, and macrophages, are reported to be primarily responsible for the synthesis of these SPMs [24,64,65] (Figure 1) [66].

### 3.2. SPMs: Resolution Function

The mechanisms involved in acute inflammatory conditions are critical for the restoration of tissue homeostasis [19,20]. The acute inflammatory response can be divided into two general phases: initiation of acute inflammation and resolution [23]. Initiation is marked by tissue edema, resulting from increased blood flow and vessel dilation that allows for the migration of leukocytes from the post-capillary lumen to the interstitial space [17,18]. This process is mediated by proinflammatory lipid mediators—namely, leukotrienes (LTs) and prostaglandins (PGs)—derived from the omega-6 fatty acid AA [28,67]. The initial recruitment of neutrophils (polymorphonuclear leukocytes (PMNs)) is followed by the recruitment of monocytes/macrophages from the blood and into the affected tissue [67,68]. In contained inflammatory exudates, coordinated lipid mediator class switching occurs in the course of acute inflammation and resolution (Figure 2) [23,69,70]. AA-derived LXA4 is the first PUFA-derived mediator found to have anti-inflammatory and pro-resolving activities [71,72]. Platelet-leukocyte interaction leads to the formation of LXA4 and LXB4, which stimulates the lipid signaling class switch by blocking the further recruitment of polymorphonuclear cells from post-capillary venules [73]. Once the noxious materials are removed via phagocytosis, the inflammatory reaction must be resolved to maintain homeostasis [70,74]. The resolution of acute inflammation is an active process that is controlled by SPMs [39,75]. These SPMs lead to the recovery of homeostasis by blocking leukocyte trafficking to the inflamed site, reversing vasodilation and vascular permeability, and promoting the clearance of inflammatory cells, exudates, and tissue debris [76,77]. In the lipid class-switch process, SPMs share similar biological functions, limiting neutrophil infiltration, shifting cytokine profiles from pro- to anti-inflammatory, and promoting macrophage phagocytosis [78].

## 4. MaRs

### 4.1. Biosynthesis of MaRs

MaRs (from macrophage mediator in resolving inflammation) are a fourth family of DHA-derived SPMs [79]. In macrophages, MaR-1 biosynthesis is initiated by 12-LOX from DHA, producing 14S-hydroperoxydocosa-4Z,7Z,10Z,12E,16Z,19Z-hexaenoic acid—the hydroperoxy intermediate—which undergoes further conversion via enzymatic 13(14)-epoxidation. This epoxide intermediate is hydrolyzed enzymatically via an acid-catalyzed nucleophilic attack by water at carbon 7, resulting in the introduction of a hydroxyl group at that position and a double bond rearrangement to form the stereochemistry of bioactive MaR1, which has potent pro-resolution properties. The 13S, 14S-epoxy-MaR intermediate is also the precursor of MaR-2 (13R, 14S-dihydroxy-4Z,7Z,9E,11Z,16Z,19Z-DHA). This product of DHA biosynthesis by 12-LOX produces the 14S-hydroperoxide that is converted to the 13S, 14S-epoxy-MaR and finally converted by a soluble epoxide hydrolase into MaR-2 (Figure 3A) [49].

### 4.2. Biosynthesis of MaR Conjugates in Tissue Regeneration (MCTRs)

Macrophages also produce a family of bioactive peptide-conjugated mediators called MCTRs [80]. MCTR compounds are produced from the 13(S), 14S-epoxide MaR intermediate during MaR biosynthesis. This epoxide intermediate is enzymatically converted to an MCTR. DHA is converted by 12-LOX into 13,14-epoxy-maresin (an intermediate of MaR-1 and MaR-2) that can be directly conjugated at C13 to glutathione by LTC4 synthase, yielding MaR conjugated in tissue regeneration 1(MCTR 1). MCTR 1 (13R-glutathionyl, 14S-hydroxy-4Z,7Z,9E,11E,13R,14S,16Z,19Z-DHA)—the first cysteinyl-SPM to be identified—is synthesized in the presence of leukotriene C4 (LTC4) synthase and γ-glutamyltransferase-μ4 in human macrophages. γ-Glutamyl transferase is involved in the conversion of MCTR-1 to MCTR2 (13R-cysteinylglycinyl, 14S-hydroxy-4Z,7Z,9E,11E,13R,14S,16Z,19Z-DHA) and MCTR-3 (13R-cysteinyl,14S-hydroxy-4Z,7Z,9E,11E,13R,14S,16Z,19Z-DHA) (Figure 3A) [34,39].

### 4.3. Function of MaRs and MCTRs

13S, 14S-epoxy-DHA(eMaR) stimulates the conversion of the M1 macrophage phenotype to M2 and blocks LTA4 hydrolase [81]. MaR1 possesses potent pro-resolving, antinociceptive, tissue regenerative, antiaggregant, and vasculoprotective functions. Recently, MaR-2 was reported to have powerful bioregulatory effects. Similarly, MCTRs act as tissue protective and regenerative agents, with anti-inflammatory and pro-resolving properties [25]. Thus, MaRs and MCTRs are regulated during acute self-limited infectious-inflammation and possess many attributes that contribute to host defense, tissue regeneration, organ protection, and pain modulation [25,34,39,49,72].

## 5. Macrophages

### 5.1. Macrophage Origin, Polarization, and Function

Naïve macrophages are widely distributed in all tissues via circulation through the bloodstream [82]. These cells remove apoptotic cells and foreign material via phagocytosis and participate in various processes, such as wound healing and tissue repair [83]. Macrophages are derived from bone marrow hematopoietic stem cells [84]. When stimulated by cytokines, bone marrow-derived macrophages develop into monocytes that then differentiate into pre-macrophages [35,85]. Finally, they become mature macrophages that can be released into the bloodstream [35,86]. Macrophages respond to current conditions to form a heterogeneous cell population [87]. Under the influence of various stimuli, they usually differentiate into one of two phenotypes (polarization) [88]: proinflammatory type (M1) and anti-inflammatory or reparative type (M2) [89]. M1 macrophages are proinflammatory and secrete cytokines, while M2 macrophages are anti-inflammatory and promote tissue repair to resolve inflammation [87]. M2 macrophages can be further sub-classified as M2a, M2b, M2c, and M2d based on transcriptional changes that result from exposure to different stimuli [90]. Lipopolysaccharides (LPS), TNF-α, and interferon gamma (IFN)-γ are used to convert macrophages to M1, IL-4 and IL-13 to M2a, immune complexes (IC) and toll-like receptors (TLR) to M2b, IL-10 to M2c, and adenosine A2A receptor (A2AR) agonist to M2d (Figure 3B) [29]. Therefore, stimulus-dependent polarization controls the specific functions and phenotypes of macrophages.

### 5.2. Relationships between Macrophages and MaRs

Macrophages participate in the biosynthesis of SPMs with both anti-inflammatory and pro-resolving properties [34]. SPMs are enzymatically biosynthesized from essential fatty acids with different stereochemistry [74,80,91]. MaRs—a new family of macrophage-derived mediators—are synthesized from DHA by macrophages and are potent in the resolution of inflammation [33]. Most importantly, MaR1 directly enhances neutrophil activation and the switch from macrophage M1 to M2 phenotype [92], both of which promotes anti-inflammatory and pro-resolving actions, inhibiting neutrophil infiltration and stimulating macrophage phagocytosis and efferocytosis to enhance the clearance of inflammation without affecting the innate response [27,49].

## 6. Role of MaRs in Inflammation Resolution

The level of this potent leukocyte agonist decreases in the later stages of the self-limited inflammatory response [93]. It is possible that other signals regulate leukocyte responses to promote tissue repair and regeneration. Given the pivotal roles of chemical signals in infections, it has been revealed that new mediators, within self-resolving infections, can regulate tissue repair and regeneration without immune suppression [34]. MaR-1 exhibits potent pro-resolving and tissue regenerative activity and is involved in self-limited infections that regulate tissue regeneration [34,94]. New pathways and mediators in planaria promote recovery and regeneration during infection [27]. The recruitment of leukocytes into the lesioned spinal cord is regulated by various proinflammatory mediators [95,96]. Cytokines mediate inflammation by acting on specific receptors that activate different intracellular inflammatory cascades [97]. Additionally, MaR-1 downregulates cytokine expression in mouse models of colitis and acute respiratory distress syndrome [41,49,92]. However, little is known about the intracellular cascades regulated by MaR-1.

Although IL-10 has anti-inflammatory properties, its contribution to the healing process is not fully established. In a recent report, MaR1 increased the levels of IL-10 postoperatively for 14 days [98]. Notably, MaR-1 has been reported to interact with stem cells to reduce chronic inflammation and improve wound healing following SCI [77,99]. Interestingly, macrophages incubated with MaR-1 are polarized toward an anti-inflammatory phenotype and increase MRC1 mRNA expression (an M2 macrophage phenotype marker), implying a possible role of MaR-1 in M2 macrophage polarization [42]. TNF-α is one of the earliest cytokines to appear following tissue damage and is associated with the production of many cytokines, including IL-1β and IL-6. MaR-1 attenuates the release of proinflammatory cytokines and TNF-α in macrophages [49,100]. In addition, intracellular adhesion molecule 1 (ICAM-1) is an epithelial PMN ligand that promotes neutrophil migration through epithelial cells during inflammation [92]. MaR-1 inhibits the 10-fold upregulation of ICAM-1, suggesting that it contributes to the resolution of inflammation by affecting neutrophil clearance and efferocytosis. Another possible avenue for treatment with MaRs is motor neuron disease, a fatal neurodegenerative disease that causes loss of motor neuron function and progressive degeneration [101]. However, the molecular mechanisms of motor neuron degeneration in amyotrophic lateral sclerosis (ALS) are not yet full known. Many pathogenic changes occur in the affected motor neurons, including mitochondrial dysfunction, hyper excitability, glutamate excitotoxicity, and nitroxidative stress [101]. Superoxide dismutase 1 (SOD1) G93A and transactivation response DNA-binding protein (TDP)-43A315T cause oxidative stress, endoplasmic reticulum (ER) stress, and inflammation. MaR-1 possesses neuroprotective effects against stress-induced cell death induced by various factors, such as SOD1 G93AA315T and TDP-43A315T, inhibiting NF-κB activation [102]. Therefore, MaR-1 may also contribute to treatment options for motor neuron diseases, such as ALS and SMA (spinal muscular atrophy).

## 7. Role of MaRs and Macrophages in Inflammation Resolution

Macrophages are involved in the processes of homeostasis, tissue repair, and regeneration [29]. They are recruited to damaged nerve sites through the activation of M1 and M2 subtypes [29]: M1 macrophages exhibit a proinflammatory profile and mediate cytotoxic actions; and M2 macrophages have anti-inflammatory effects and promote tissue healing and recovery [31,32]. The balance of M1 and M2 macrophages regulates early events in local inflammation [103]. In this process, cytokine contributes to the recruitment of M1 macrophages [104]. It releases other proinflammatory cytokines, such as TNF-α, IL-1α, and IL-β, to promote further tissue damage [105]. To control this process, activated M2 macrophages release anti-inflammatory cytokines, IL-10 and TGF-β, which mediate tissue regeneration and inhibition of the proinflammatory function [106]. Also, an increased ratio of M2 macrophages significantly enhances nerve regeneration and wound healing [107]. DHA plays important roles in peripheral organs, as well as in the central nervous system, and is the precursor of various molecules that regulate the resolution of inflammation [50,108]. Macrophages derived from mice deficient of Elov12 (Elovl2-/-)—the main enzyme for DHA synthesis—demonstrate an increased expression of M1-like markers (iNOS and CD86), whereas M2 macrophages downregulate M2-like markers such as CD206 [50]. Similarly, the impairment of systemic DHA synthesis in activated macrophages results in an alteration of M1/M2 macrophages, supporting the important role played by DHA in regulating the balance between pro- and anti-inflammatory processes. Inflammation resolution is an active and highly regulated inflammatory process that is necessary to prevent the transition into chronic inflammation with the spread of tissue injury or exacerbated scarring [13]. However, differential leukocyte subpopulations reduce or otherwise impair the ability to resolve inflammation at the lesion site after acute experimental spinal cord injury (SCI) [95,103]. For example, after SCI, M2 macrophage function is shown to be defective in resolving inflammation, impairing tissue remodeling and healing [77]. Macrophages in SCI are not defined within the M1-M2 dichotomy. MaR1 is effective in enhancing several stages of inflammation resolution after SCI through the downregulation of cytokines, reduction of neutrophils and macrophages, shift in macrophage phenotype, and stimulation of macrophage phagocytosis [77]. Treatment with MaR1 after SCI reportedly enhances neutrophil clearance and reduces macrophage accumulation in the lesioned tissue [77]. Inappropriate biosynthesis of SPMs after SCI interferes with the resolution of inflammation and contributes to the pathophysiology of SCI. Abnormal production of SPMs is also reported in the cerebrospinal fluid (CSF) of patients with Alzheimer’s disease and multiple sclerosis [68,75]. Thus, the potential function of MaRs can be confirmed by their immunoresolvent effects on the phagocytosis of macrophages and their inhibitory functions on cytokine levels and inflammatory signaling pathways [77]. Chronic inflammation is the basis of the common pathology of age-related diseases, such as cardiovascular disease, diabetes, and Alzheimer’s disease [2], involving alterations to the immune system that promote chronic inflammation. Macrophages are important in these age-associated changes that cause chronic inflammatory diseases [72]. Recent studies have shown that aging impairs macrophage phagocytosis, resulting in a failure to resolve damage-associated molecular patterns in aged animals [2,45,51].

## 8. Role of MaRs in Resolution of Inflammatory Pain

MaR1 not only regulates the resolution of inflammation, but also plays a powerful role in preventing hyperalgesia sensitivity in inflammatory- and chemotherapy-induced chronic inflammatory pain [23,27]. MaR1 dramatically reduces vincristine-initiated neuropathic pain in a cancer chemotherapy model [27] and in temporomandibular joint pain [26]. In addition, MaR1 has played an important role in the prevention of postoperative pain in orthopedic surgery models [24]. Postoperative pain management with MaR1 may help control the onset of neuroinflammation. Acute perioperative treatment with MaR1 delayed the development of mechanical and cold allodynia. Moreover, MaR1 has been shown to induce analgesia by regulating transient receptor potential vanilloid 1 (TRPV1) currents in neurons. [27]. Intrathecal treatment with MaR1 reduces inflammatory pain with a long-lasting analgesic profile through the inhibition of astrocytic and microglial activation [109].

In the periphery, various cytokines contribute to neutrophil recruitment into the tissue and, consequently, an increase in inflammatory processes and pain [28,110]. Intrathecal MaR1 treatment reduces recruitment and leukocyte count [94,100]. In addition, MaR1 reduces CFA-induced mRNA expression of Nav1.8 and Trpv1 channels [26]. MaR1 likely controls TRPV1 expression in DRG neurons during inflammation. Thus, targeting these channels is effective for reducing inflammatory pain [111,112]. Under noxious stimuli, nociceptor neurons release neuropeptides, such as CGRP and substance P, which control the recruitment of immune cells to the inflamed tissue [113,114]. MaR1 reduced the release of CGRP from DRG neurons, indicating a possible mechanism by which MaR1 reduces inflammatory pain through reduced recruitment of neutrophils and macrophages. In the spinal cord, TNF-α and IL-1β contribute to spinal cord plasticity and, hence, central sensitization [97]. Cytokines improve the amplitude of AMPA- and glutamate-induced excitatory currents [97]. Indeed, the CFA model induces central sensitization with stronger activation of astrocytes when compared to microglia [111,115,116]. Central sensitization has been recognized as the main cause of pathological pain, resulting in plastic changes in the CNS [117]. Intrathecal treatment with MaR1 reduced CFA-induced astrocyte and microglial activation and decreased activation by TNF-α, IL-1β, and NF-κB [26]. In addition, the interaction between glial cells and nociceptor neurons has been linked to these plastic changes in the spinal cord [26]. MaR1 reduces glial cell activation and blocks capsaicin-induced TRPV1 calcium influx as well as spontaneous EPSC frequency [26]. Thus, MaR1 can reduce spinal cord plastic changes and inhibit central sensitization via presynaptic and postsynaptic mechanisms [26]. MCTRs also rescue *Escherichia coli* infection-mediated delays in tissue regeneration of planaria, in addition to protecting mice from second-organ reflow injury and promoting repair by limiting neutrophil infiltration [41]. Furthermore, each MCTR promotes the resolution of *E. coli* infections by increasing bacterial phagocytosis and limiting neutrophil infiltration [25]. Phagocytosis is a main means by which macrophages resolve inflammation [2]. Increasing evidence suggests that MaRs produce potent anti-inflammatory and pro-resolution profiles, partially by enhancing macrophage activity [17,93]. However, it remains unclear how MaRs regulate macrophage phagocytosis. 

## 9. Conclusions

MaRs belong to the most recently uncovered family of anti-inflammatory lipid mediators with pro-resolving activity in the amelioration of inflammation. The activation of MaRs in macrophages enhances phagocytosis and helps to reverse inflammatory pain by shifting cytokine release to an anti-inflammatory state. However, M1/M2 macrophage imbalance, reduced SPM formation, impaired synthesis of DHA, and aging reduce and generally impair the resolution of inflammation under pathological conditions. MaRs have been shown to alleviate this deficiency by reversing and improving the function of macrophages (Figure 4). However, the specific receptors and signaling mechanisms involved in the ability of MaRs to resolve inflammation must be investigated in order to fully establish their important role in the treatment of inflammation in various diseases.

## Figures and Tables

**Figure 1 ijms-20-05849-f001:**
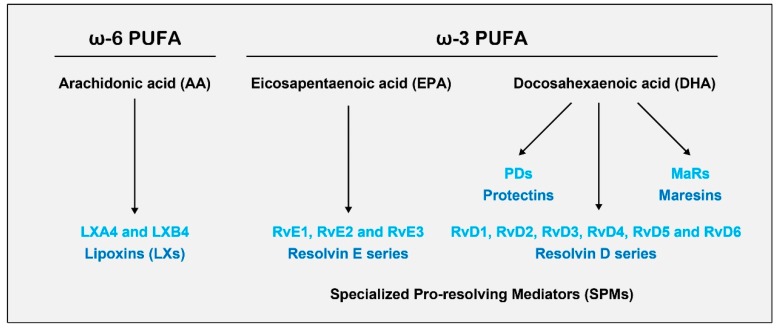
Biosynthesis of specialized pro-resolving mediators (SPMs) derived from omega-3 and omega-6 polyunsaturated fatty acids (PUFAs).

**Figure 2 ijms-20-05849-f002:**
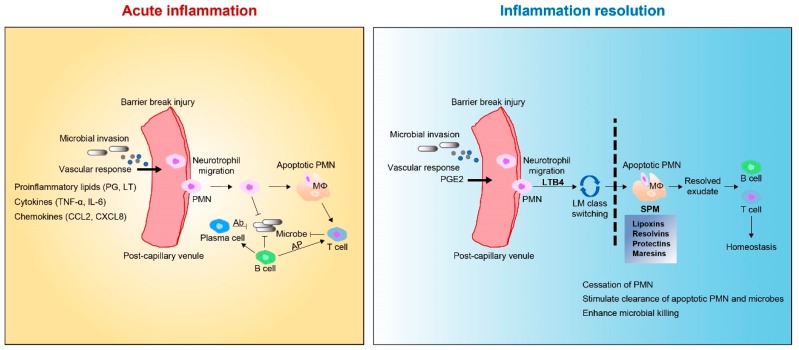
The outcome of acute inflammation and resolution. Under stimulation of injury or infection, release of proinflammatory lipids (prostaglandin (PG), leukotriene (LT)), chemokines (C-C motif chemokine ligand 2 (CCL2), C-X-C motif ligand 8 (CXCL8)), and cytokines (tumor necrosis factor-α (TNF-α), interleukin (IL)-6) induce the recruitment of neutrophils. Other immune cells (macrophages, B cells, and T cells) also participate in the process. Macrophages directly phagocytize organisms and apoptotic neutrophils, while B cells are converted into plasma cells to kill organisms through secreted antibodies, referred to as antibody-dependent cell-mediated cytotoxicity. Macrophages and B cells activate T cells via antigen cross presentation (AP). PGE2 leads to vasodilation and LTB4 stimulates PMN influx into the inflammatory locus. Subsequently, lipid mediator (LM) class switching converts proinflammatory signals into pro-resolving signals and triggers resolution. SPMs restrict excessive PMN influx to the injury site, enhance efferocytosis, and stimulate pro-resolving signals.

**Figure 3 ijms-20-05849-f003:**
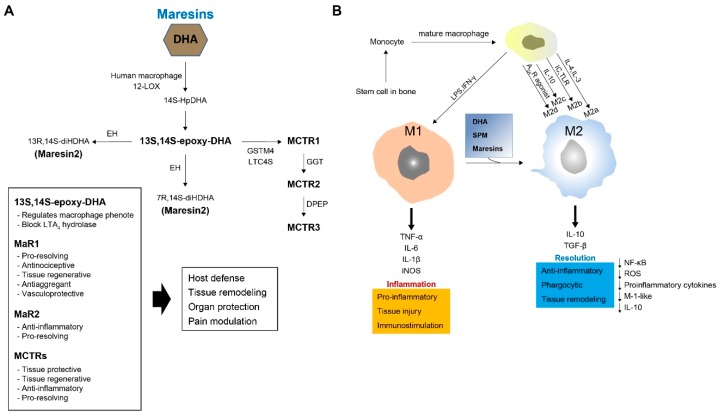
Synthesis and function of maresins (MaRs) and macrophages. (**A**) MaRs and MaR conjugates in tissue regeneration (MCTRs) biosynthesis. Human macrophage 12-LOX converts DHA to the 13S,14S-epoxy-maresin intermediate and hydrolase or soluble epoxide hydrolase is converted to MaR1 and MaR2, respectively. The MCTR biosynthetic pathway is initiated by lipoxygenation of 14S-HpDHA, converted by lipoxygenase activity to the 13S,14S-epoxy-maresin intermediate. MCTR1 is catalyzed by glutathione s-transferase mu4 (GSTM4) and/or leukotriene C4 synthase (LTC4S). MCTR1 is converted by gamma-glutamyl transferase (GGT) to MCTR2, which then acts as a substrate for conversion by dipeptidase (DPEP) to MCTR3. (**B**) M1 and M2 polarization of macrophages. Bone marrow-derived macrophages differentiate into mononuclear cells and gradually become mature macrophages that can be released into circulation. IFN-γ, TNF-α, and LPS stimulate macrophages into M1, IL-4 and IL-13 into M2a, IC and TLR into M2b, and IL-10 into M2c; A2AR agonist stimulates them into M2d. M1 macrophages induce a proinflammatory response, whereas M2 macrophages induce an anti-inflammatory response. M1 macrophages can also differentiate into M2 macrophages through local cues. The M1 phenotype is proinflammatory, phagocytic, and bactericidal, while the M2 macrophages act to switch off inflammation. IFN-γ: interferon gamma; TNF-α: tumor necrosis factor alpha; LPS: lipopolysaccharides; IC: immune complexes; TLR: toll-like receptor; A2AR: adenosine A2A receptor; IL: interleukin; IL-1R: IL-1 receptor.

**Figure 4 ijms-20-05849-f004:**
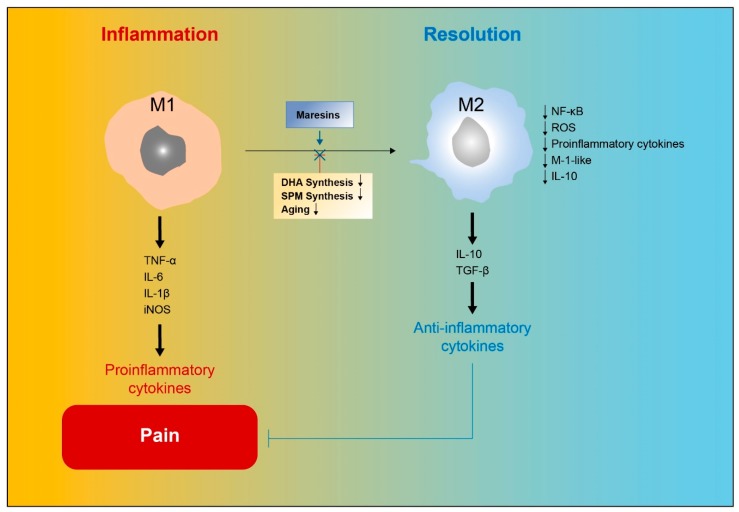
Maresins regulate macrophage phenotype and resolution of inflammatory pain. Maresins (MaRs) improve M2 macrophage function, shifting cytokine release to an anti-inflammatory profile and thereby facilitating the resolution of inflammatory pain.

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
