# Peer review of "The Role of Maresins in Inflammatory Pain: Function of Macrophages in Wound Regeneration"

_ijms, 2019, doi:10.3390/ijms20235849_

Round 1
Reviewer 1 Report
It's a good idea to review and understand the underlying mechanisms of pain and inflammatory conditions, in which the changes of macrophage phenotype and functions are play an important role.
In this review paper, the author believe that maresins (MaRs) facilitated the resolution of inflammatory pain by up regulating M2 function and increasing anti-inflammatory cytokines. I only have a few recommendation below
It'd be better to change the title as it not fit with wound repair and regeneration, a bit confused. would like to upload a higher resolution figure
Reviewer 2 Report
The review is complex in content but very interesting.
Introduction: add one / two general paragraphs explaining overall what occurs when there is a tissue injury (from lesion to regeneration)
Figure 1: what is PUFA?
Add a chapter that refers exclusively to wound repair cutaneous
Author Response
We thank reviewer #2 for the helpful comments that further strengthened the manuscript. We agree with three points of reviewer #2 and have revised as follows:
Point 1: Introduction: add one / two general paragraphs explaining overall what occurs when there is a tissue injury (from lesion to regeneration)
Response 1: In the introduction, we added one general paragraph explaining a tissue injury (from lesion to regeneration):
Inflammation is an immune response to harmful stimuli, including pathogens, damaged cells, toxic compounds, surgery, or irradiation [1]. Inflammation is characterized by swelling, heat, pain, redness, and loss of tissue function, which is caused by local immune, vascular, and inflammatory cell responses to infection or injury [2]. Inflammatory processes that include changes in vascular permeability, recruitment and accumulation of leukocytes, and release of inflammatory mediators, are important in the regeneration of injured tissues [3]. Therefore, inflammation is an essential defense mechanism for preserving health. A weak inflammatory response can lead to tissue destruction by harmful stimuli, while chronic unresolved inflammation may culminate in various pathological conditions, including cancer, fibrosis, and pain [4]. Wound regeneration promotes resolution of inflammation by restoring barrier function [5]. Neutrophils are the first circulating inflammatory cells to be recruited to the wound site [6]. Clinical observations demonstrating that leukocyte recruitment disorders and reduced neutrophil infiltration are associated with delayed wound healing indicate the importance of neutrophils for efficient wound repair [7]. Recent studies have shown that macrophages exhibit different functions during the immune response, with pro-inflammatory signaling occurring during the early stages of inflammation and, once inflammation is resolved, promotion of tissue regeneration at late stages [8, 9].
Point 2: Figure 1: what is PUFA?
Response 2: We added the full name of PUFA to the legend of Figure 1: “Polyunsaturated fatty acids (PUFAs)”
Point 3: Add a chapter that refers exclusively to wound repair cutaneous
Response 3: We added a chapter to the manuscript explaining the mechanism of cutaneous wound repair.
1. Tissue inflammation and regeneration
1.1 Tissue inflammation
The inflammatory response after tissue damage is an important biological process that is essential for the survival of living organisms [1]. When tissues are damaged by infection, exposure to toxins, or mechanical damage, an inflammatory response is induced by damage-associated molecular patterns (DAMPs) and pathogen-associated molecular patterns (PRR) released by dead cells and invading organisms [10]. These molecules provoke a complex inflammatory response characterized by the recruitment, proliferation, and activation of various hematopoietic and non-hematopoietic cells, including neutrophils, macrophages, innate lymphoid cells, natural killer cells, B cells, T cells, fibroblasts, epithelial cells,
endothelial cells, and stem cells, which together constitute the cellular response that orchestrates tissue repair [11].
1.2 Tissue regeneration
When the wound healing reaction is well organized and controlled, the inflammatory response is quickly resolved, and normal tissue structure is restored [12]. However, if the wound healing response is chronic or becomes dysregulated, it can lead to the development of pathological fibrosis or scars, which impairs normal tissue function and may ultimately lead to organ failure and death [11]. Therefore, the wound-healing reaction must be strictly regulated. Biological processes involved in cutaneous wound healing include infiltration of inflammatory cells, fibroblast repopulation and new vessel formation, keratinocyte migration, and proliferation [13]. Although many cells are involved in tissue repair, macrophages exhibit significant regulatory activity at all repair and fibrosis stages and are critically involved in normal tissue homeostasis [14]. It is clear that monocytes and macrophages play more complex roles in tissue repair and in contributing to fibrosis and tissue regeneration [8]. Macrophages are an important source of chemokines, matrix metalloproteinases (MMPs), and other inflammatory mediators that induce early cellular response after injury [15]. Indeed, when macrophages are depleted soon after injury, the inflammatory response is often greatly reduced [8]. However, their removal can also reduce wound debris and cause less efficient repair and regeneration [11]. After the initial inflammatory phase subsides, the main macrophage population develops a wound healing phenotype characterized by the production of numerous growth factors, such as transforming growth factor beta 1 (TGF-β1), platelet-derived growth factor (PDGF), vascular endothelial growth factor alpha (VEGF-α), and insulin-like growth factor 1 (IGF-1) [5].
Round 2
Reviewer 2 Report
The observations were received, however I would add the following bibliographical reference which underlines the important action of platelet degranulation in the healing processes

Author Response
Response to Reviewer 2 Comments :
We thank reviewer #2 for the helpful comments that further strengthened the manuscript. We agree with the point of reviewer #2 and have revised as follows:
Point 1: The observations were received, however I would add the following bibliographical reference which underlines the important action of platelet degranulation in the healing processes.
Response 1-1: As you suggested, we have added three new references to the end of the paragraph.
These molecules provoke a complex inflammatory response characterized by the recruitment, proliferation, and activation of various hematopoietic and non-hematopoietic cells, including neutrophils, macrophages, innate lymphoid cells, natural killer cells, B cells, T cells, fibroblasts, epithelial cells, endothelial cells, and stem cells, which together constitute the cellular response that orchestrates tissue repair (Reference 54, 55 and 56)
54. Nicoletti, G.; Saler, M.; Villani, L.; Rumolo, A.; Tresoldi, M. M.; Faga, A., Platelet Rich Plasma Enhancement of Skin Regeneration in an ex-vivo Human Experimental Model. Front Bioeng Biotechnol 2019, 7, 2.
55. Tsepkolenko, A.; Tsepkolenko, V.; Dash, S.; Mishra, A.; Bader, A.; Melerzanov, A.; Giri, S., The regenerative potential of skin and the immune system. Clin Cosmet Investig Dermatol 2019, 12, 519-532.
56. Wynn, T. A.; Vannella, K. M., Macrophages in Tissue Repair, Regeneration, and Fibrosis. Immunity 2016, 44, (3), 450-462.
Response 1-2: As you suggested, we also have added three new references to the end of the paragraph.
Biological processes involved in cutaneous wound healing include infiltration of inflammatory cells, fibroblast repopulation and new vessel formation, keratinocyte migration, and proliferation (Reference 54, 58 and 59)
54. Nicoletti, G.; Saler, M.; Villani, L.; Rumolo, A.; Tresoldi, M. M.; Faga, A., Platelet Rich Plasma Enhancement of Skin Regeneration in an ex-vivo Human Experimental Model. Front Bioeng Biotechnol 2019, 7, 2.
58. Serra, M. B.; Barroso, W. A.; da Silva, N. N.; Silva, S. D. N.; Borges, A. C. R.; Abreu, I. C.; Borges, M., From Inflammation to Current and Alternative Therapies Involved in Wound Healing. Int J Inflam 2017, 2017, 3406215.
59. Ridiandries, A.; Tan, J. T. M.; Bursill, C. A., The Role of Chemokines in Wound Healing. Int J Mol Sci 2018, 19, (10).